# Endoplasmic Reticulum Associated Degradation of Spinocerebellar Ataxia-Related CD10 Cysteine Mutant

**DOI:** 10.3390/ijms21124237

**Published:** 2020-06-14

**Authors:** Mai Kanuka, Fuka Ouchi, Nagisa Kato, Riko Katsuki, Saori Ito, Kohta Miura, Masaki Hikida, Taku Tamura

**Affiliations:** 1Department of Life Science, Graduate school of Engineering and Resource, Akita University, Akita 010-8502, Japan; mai20878@icloud.com (M.K.); krh.wtsn.mad.921@gmail.com (N.K.); 430t0mat0@gmail.com (R.K.); saorin.23m@gmail.com (S.I.); m8020028@s.akita-u.ac.jp (K.M.); hikida@gipc.akita-u.ac.jp (M.H.); 2Department of Life Science, Faculty of Engineering Science, Akita University, Akita 010-8502, Japan; ouchifuka@gmail.com

**Keywords:** spinocerebellar ataxia, CD10, disulfide bond, endoplasmic reticulum-associated degradation (ERAD), ER quality control

## Abstract

Spinocerebellar ataxia (SCA) is one of the most severe neurodegenerative diseases and is often associated with misfolded protein aggregates derived from the genetic mutation of related genes. Recently, mutations in CD10 such as C143Y have been identified as SCA type 43. CD10, also known as neprilysin or neuroendopeptidase, digests functional neuropeptides, such as amyloid beta, in the extracellular region. In this study, we explored the cellular behavior of CD10 C143Y to gain an insight into the functional relationship of the mutation and SCA pathology. We found that wild-type CD10 is expressed on the plasma membrane and exhibits endopeptidase activity in a cultured cell line. CD10 C143Y, however, forms a disulfide bond-mediated oligomer that does not appear by the wild-type CD10. Furthermore, the CD10 C143Y mutant was retained in the endoplasmic reticulum (ER) by the molecular chaperone BiP and was degraded through the ER-associated degradation (ERAD) process, in which representative ERAD factors including EDEM1, SEL1L, and Hrd1 participate in the degradation. Suppression of CD10 C143Y ERAD recovers intracellular transport but not enzymatic activity. Our results indicate that the C143Y mutation in CD10 negatively affects protein maturation and results in ER retention and following ERAD. These findings provide beneficial insight into SCA type 43 pathology.

## 1. Introduction

Production of functional proteins in the secretory pathway is an important process to maintain inter- and intracellular homeostasis. Protein misfolding occurs due to various conditions, such as cellular crowd, genetic mutation, or environmental stress, eventually producing functionally disordered proteins. Normally, such undesirable proteins are recognized by the proteolytic machinery, including molecular chaperones and post-translational modifying enzymes, and are sent to the cellular degradation systems. However, exceeding the degradation capacity increases unstable protein levels and induces toxic protein aggregation. Typically, long-term accumulation of aggregates leads to conformational diseases, including neurodegenerative degeneration [1].

Spinocerebellar ataxia (SCA) is a progressive neurodegenerative disorder exhibiting pathological and clinical features, such as cerebellar cell death and movement disorders [2]. SCA consists of a large heterogeneous group of genetic disorders that are inherited in an autosomal dominant fashion. More than 40 types of SCA have been characterized so far and are closely associated with the toxic misfolded protein aggregates [3]. Similar to other conformational diseases such as Alzheimer’s disease, long-term accumulation of aberrant SCA-related proteins in the neurons causes dysfunctions and disorders. Understanding the molecular mechanism of protein misfolding and subsequent aggregation is important for effective treatment of SCA.

In mammalian cells, functional secretory soluble proteins and membrane proteins are initially assembled in the endoplasmic reticulum (ER). During the maturation process, newly synthesized proteins fold into a functional structure with the assistance of molecular chaperones and co- and post-translational modification by the resident N-glycan modification enzymes and protein disulfide isomerase family. When protein folding in the ER fails, misfolded polypeptides are isolated from the folding machinery and are then sent to the degradation process called ER-associated degradation (ERAD). Recent reports have unveiled the role of secretory and membrane proteostasis in cell homeostasis, the failure of which is closely associated with conformational diseases [4].

CD10 (also known as neprilysin or neutral endopeptidase 24.11) is expressed on the cell surface and regulates development of various organs and tissues including the B cell, stem cells, and lung by its endopeptidase activity [5,6,7]. CD10 is a type II transmembrane protein containing six intramolecular disulfide bonds and three N-linked glycans (Figure 1A). CD10 is located at the raft domain of the plasma membrane, which is the site for signal transduction where related proteins are clustered, and targets various physiologically active peptides, including amyloid beta (Aβ), neurotensin, and oxytocin [5]. The clinically most important function of CD10 is to digest the Aβ peptide, which is the hallmark of Alzheimer’s disease. Aβ is generated from the processing of amyloid precursor protein by beta or gamma secretases at the cell surface [1]. CD10 has the ability to digest Aβ, irrespective of the monomeric or oligomeric (cytotoxic) form in cultured cells [8]. Moreover, CD10 has been used as a cancer marker that regulates stem cell growth [5,9]. Thus, the importance of CD10 in the field of clinical medicine has been increased. However, biosynthetic pathways of CD10, including the protein maturation process in the ER and disposal process of misfolded CD10, are largely unknown.

The CD10 mutation has been observed in a patient with SCA subtype 43, wherein the cysteine residue at 143 is replaced by tyrosine [10]. As Cys143 forms an intramolecular disulfide bond with Cys411, the mutation in Cys143 residue possibly causes incorrect disulfide bond formation that may affect protein folding and enzymatic activity. Functional details of the gene product of the SCA-related CD10 mutation remain to be elucidated. In this study, we aimed to characterize this neurodegenerative disease mutant with regard to cellular transport, ER quality control processes, and enzyme activity. We found that the CD10 C143Y mutant forms an aberrant disulfide bond-mediated oligomer, and that the ER protein quality control system suppresses its transport to the plasma membrane. Furthermore, the ER-retained CD10 C143Y mutant is degraded via the EDEM1, SEL1L, and Hrd1-mediated ERAD pathway. Our results indicated that CD10 C143Y is a novel ERAD substrate and provides the molecular basis for understanding SCA due to cysteine mutation in CD10.

## 2. Results

### 2.1. SCA-Related Cysteine Mutation in CD10 Causes Aberrant Disulfide Bond Formation and Changes in Raft Recruitment

Recently, CD10 has been identified as a causative gene for SCA [10]. To explore the outcome of cysteine mutation in CD10, we developed expression vectors of WT and the SCA mutant CD10 with a point mutation C143Y (Figure 1A). As a first step to understanding the influence of the C143Y mutation in SCA 43 development, functional characterization of CD10, including protein maturation, intracellular transport, and enzyme activity, was carried out using HeLa cells because endogenous CD10 protein expression was hardly detected by our experimental system (Appendix A). Of note, we used mouse CD10 cDNA, which has a 97.7% similarity to that of human CD10. Structural features, including the transmembrane region, catalytic domain, and cysteine positions, were mostly conserved.

Previous reports showed that endogenous and exogenously expressed WT CD10 were located at the raft domain on the cell surface [11,12]. To monitor CD10 intracellular transport and raft incorporation in our experimental system, cell lysates from the transfected cells were fractionated into 1% Triton X-100-soluble (non-raft) and -insoluble (raft) fractions. Additionally, cell lysates were denatured in the presence or absence of a reducing agent, dithiothreitol, to analyze the status of disulfide bond formation. Under reducing conditions, WT CD10 and C143Y were detected as a single band by immunoblotting (Figure 1B). In agreement with a previous report [11], the majority of WT CD10 was detected in the detergent-insoluble, pellet fraction (P) (Figure 1B, lane 2), indicating that exogenously expressed WT CD10 in HeLa cells was well-folded and recruited to the raft domain of the plasma membrane. However, CD10 C143Y was found in low quantity in P (Figure 1B, compare lanes 3 and 4). Of note, faster migration of CD10 C143Y than that of WT CD10 (compare lane 2 with 3 of Figure 1B) was observed probably due to the lack of N-glycan modification at the Golgi, which was confirmed by Endo H or PNGase F treatment (Figure 2A). Quantification of detergent-insolubility by the quantification of WT CD10 and C143Y revealed that approximately 70% of WT CD10 was detected in the detergent-insoluble fraction, whereas approximately 90% of C143Y appeared in the soluble fractions (Figure 1C). These results suggest that cysteine mutation at position 143 of CD10 affects cellular trafficking and raft recruitment.

To examine the structural influence of the C143Y mutation on disulfide bond formation of CD10, cell lysates of WT CD10- and C143Y-expressing HeLa cells were analyzed by Western blotting under the non-reducing condition. WT CD10 was detected as a single band under the reducing condition (compare Figure 1B,D, lane 1–2). CD10 C143Y, however, exhibited high-molecular-weight forms (Figure 1D, lane 3–4). As the reducing condition did not yield such multiple bands, the appearance of high-molecular-weight forms under the non-reducing condition reflected the formation of several intermolecular disulfide bonds in CD10 C143Y. Collectively, these results support the observation that Cys143 of CD10 is important for disulfide bond formation and transport to the raft domain of the plasma membrane.

### 2.2. CD10 C143Y Is Retained in the ER

Next, we addressed the deficiency of intracellular transport of CD10 by the consequence of C143Y mutation. We performed endo H digestion of N-glycan to monitor the cellular localization of CD10. Endoglycosidase H (Endo H) processes only ER and the medial-Golgi form of N-glycan on the glycoproteins; therefore, this treatment reveals the cellular localization of glycoproteins by mobility shift in Western blotting. The migration remained unchanged by either mock treatment or endo H digestion in WT CD10 (compare lanes 1–2 and 3–4 in Figure 2A), indicating that N-glycans of WT CD10 exogenously expressed in HeLa cells were appropriately modified at the Golgi during transport to the cell surface. On the other hand, CD10 C143Y was completely sensitive to the endo H treatment (compare lanes 7–8 and 9–10 in Figure 2A). This mobility shift was indistinguishable from that of PNGase F digestion, which removes N-linked glycan irrespective of the cellular localization of the glycoproteins (Figure 2A, lanes 11–12). These results suggest that WT CD10 is transported to the cell surface, whereas intracellular transport of the C143Y mutant is limited to the ER and medial-Golgi region.

To ensure cellular localization of WT CD10 and C143Y mutant, we constructed CD10-mCherry fused at the C-terminus and performed live cell imaging analysis. CD10 WT-mCherry exhibited Golgi and a plasma membrane-like morphology (Figure 2B, upper panels) and no co-localization with Sec61β fused with SGFP2, which was used as the ER marker. This result is consistent with the notion that WT CD10 is detected in the detergent-insoluble fraction that corresponds to the raft domain of the cell surface and is endo H-resistant (Figure 1B and Figure 2A). However, CD10 C143Y-mCherry was hardly detected at the plasma membrane and was strongly co-localized with the ER marker (Figure 2B, lower panels). As the C143Y mutation results in disulfide bond-mediated oligomerization (Figure 1B), these results clearly demonstrate that C143Y mutation impacts the intracellular transport of CD10 by preventing ER exit.

### 2.3. BiP, an ER-Resident Hsp70 Chaperone, Interacts with CD10 C143Y but Not with WT CD10

The ER quality control system retains immature or terminally misfolded secretory and membrane proteins from subsequent vesicle transport from the ER. This retention system is executed by ER-resident chaperones that recognize the exposed hydrophobic region of the misfolded proteins [13,14]. To identify the factor that retains CD10 C143Y in the ER, CD10 C143Y-Flag was immunoisolated, and the bound proteins were analyzed.

Anti-KDEL antibody recognizes the KDEL sequence motif at the C-terminus and detects representative ER chaperones, including GRP94, BiP, calreticulin (CRT), and protein disulfide isomerase (PDI) (Figure 3A, lanes 7–9). Immunoblotting with the anti-KDEL antibody, following protein isolation of Flag-tagged CD10 using anti-Flag agarose beads, showed that BiP, an ER-resident Hsp70 homolog, binds to C143Y but not to WT CD10 (Figure 3A, lane 11 and 12). As BiP recognizes the hydrophobic moiety of misfolded proteins to prevent further aggregation [15,16], these results strongly suggest that C143Y mutation in CD10 causes protein misfolding, resulting in increased BiP binding. Interaction of CRT (a lectin-type molecular chaperone), PDI (disulfide bond formation and stabilization), and GRP94 (HSP90-like molecular chaperone) was hardly detected (Figure 3A, lane 11 and 12), suggesting that the major phenotype of C143Y mutation is the development of hydrophobicity at the molecular surface. Our results support the notion that SCA-related CD10 mutant C143Y is retained in the ER, and the molecular chaperone BiP strongly participates in this retention.

### 2.4. C143Y Mutation Destabilizes CD10

As cysteine mutation at 143 to tyrosine exerted negative effects on CD10, CD10 C143Y should have been disposed by the protein quality control system of the ER to prevent further aggregation. To analyze the CD10 C143Y quality control mechanism, transfected HeLa cells were treated with cycloheximide (CHX) to suppress novel protein synthesis, and the protein degradation pathway was examined.

The results showed that CD10 C143Y was decreased to approximately 50% by CHX-chase before 3 h, whereas α-tubulin, as the cellular control, did not change significantly (Figure 4A lanes 1–3). CD10 WT-Flag was turned over slowly in the CHX-chase assay (Appendix A), suggesting that cell surface-resident CD10 WT is more stable than C143Y.

To identify the cellular degradation pathway, transfected cells were CHX-chased for 6 h in the presence of ERAD inhibitory drugs or lysosome inhibitors. Kifunensine (KIF) inhibits ER mannosidase I and stabilizes ERAD substrate proteins [17,18]. MG132 and Clq are proteasome and lysosome inhibitors, respectively. As shown in Figure 4A,B, the degradation of CD10 C143Y was slightly recovered by ERAD inhibitors KIF and MG132 compared to that under conditions of CHX alone at 6 h (Figure 4A, lanes 3–5). The lysosome inhibitor Clq, however, showed no recovery effect on CD10 (Figure 4A, compare lanes 3 and 6), suggesting that CD10 C143Y is not transported to the lysosome for degradation, and autophagy is not involved in CD10 C143Y turnover (Figure 4A, lane 6).

### 2.5. CD10 C143Y Is Degraded through the EDEM1–SEL1L–Hrd1 Pathway of ERAD

Next, we attempted to identify the factors involved in CD10 C143Y degradation. In our effort to establish the pathway, we inspected representative ERAD factors, EDEM1, SEL1L, and Hrd1. As EDEM1 accelerates the degradation rate of different misfolded proteins in ERAD [19,20,21], we examined whether overexpression of EDEM1 promotes ERAD of CD10 C143Y. As the amount of co-expression of HA-tagged EDEM1 increased, the remaining cellular expression level of CD10 C143Y decreased (Figure 5A,B). SEL1L, a component of Hrd1 retrotranslocation machinery situated in the ER membrane, acts as an adaptor for EDEM1 (and also XTP3B and OS-9) in which EDEM1 delivers misfolded proteins to SEL1L [22,23,24]. Next, we performed an RNAi experiment to knockdown SEL1L gene expression and to verify its role in the degradation of CD10 C143Y. Compared to the negative control (luciferase; Figure 5C, lanes 1–3), RNAi to SEL1L decreased the SEL1L protein expression level (Figure 5C, lanes 4–6). In this condition, CD10 C143Y degradation was significantly delayed in CHX-chase (Figure 5C,D). These results suggest that SEL1L is required for efficient CD10 C143Y turnover.

Hrd1 is an E3 ligase that recognizes and ubiquitinates ERAD substrates for transportation from the ER to the cytoplasm [25,26]. To examine the effect of Hrd1 polyubiquitination on CD10 C143Y degradation, we exogenously expressed myc-tagged Hrd1 WT or c/s mutant, which is deficient in ubiquitin ligase activity [27]. As shown in Figure 5E,F, co-expression of WT Hrd1 reduces CD10 C143Y expression, whereas c/s mutation increases its level, suggesting that Hrd1 E3 activity is required for CD10 C143 disposal. Together, these results clearly indicate that CD10 C143Y is converted to an ERAD substrate by the cysteine mutation related to SCA and suggest that CD10 C143Y degradation is mediated by ERAD of the EDEM1–SEL1L–Hrd1 pathway.

### 2.6. CD10 C143Y Accumulates at the Aggresome during Proteasome Inhibition

To further characterize the intracellular degradation mechanism of SCA CD10 mutant, CD10 C143Y fused with mCherry at the C-terminus was tracked using live cells. HeLa cells were transfected with CD10 C143Y-mCherry and Sec61βΔTM-SGFP2, an ER organelle marker, and cells were treated with CHX and MG132 to prevent de novo protein synthesis and proteasomal degradation. For chase up to 360 min, CD10 C143Y largely appeared as a cluster juxtaposed to the nucleus, whereas Sec61βΔTM-SGFP2 showed an intact ER network (Figure 6A). Indirect immunostaining analysis showed that this structure is surrounded by intermediate vimentin filaments (Figure 6B). Misfolded ER or cytoplasmic proteins are temporarily transported to the aggresome, when the proteasome activity is downregulated or is over its capacity, to suppress the formation of cytoplasmic protein aggregates [28,29]. Our results suggested that CD10 C143Y is constantly degraded through ERAD and is recruited to the aggresome when proteasome degradation is impaired. Of note, this morphological alteration of CD10 C143Y was not due to transport from the ER to the Golgi, as there was no co-localization with GM130, a medial-Golgi marker (Appendix A). Taken together, these results indicate that CD10 C143Y is accumulated in the ER due to imperfect disulfide bond formation and is persistently degraded by the proteasome by dislocation from the ER membrane to the cytoplasm.

### 2.7. Kifunensine Suppresses CD10 C143Y Degradation and Recovers Transport to the Cell Surface but Not Endopeptidase Activity

While the gene expression level of ERAD-related factors greatly affects CD10 C143Y degradation (Figure 5), a representative ERAD inhibitor KIF did not show significant impact in the CHX-chase assay system (Figure 4). There is a possibility that the majority of CD10 C143Y found in our immunoblotting experimental condition is post-ERAD surveillance by EDEM1 as accumulation in the ER by BiP association (Figure 3A). Therefore, we speculate that suppression of ERAD of CD10 C143Y could be detected by long-term KIF treatment. In the total 48 h of transfection, incubation with KIF in the last 16 h markedly recovered CD10 C143Y protein expression level (Figure 7A,B). In this condition, cell surface expression of CD10 C143Y was also recovered (Figure 7C). These results suggest that CD10 C143Y constitutively degraded through the ERAD pathway, and KIF treatment enables us to downregulate CD10 C143Y degradation and enhances its intracellular transport to the cell surface.

As CD10 C143Y exhibits an aberrant oligomeric state containing intermolecular mixed disulfide bonds (Figure 1D), this structural alteration could affect its endopeptidase activity. We measured the enzymatic activity of WT and C143Y CD10 using the artificial substrate *Z*-ALL-pNA. Cleavage of *Z*-ALL-pNA by endopeptidase releases pNA, and the enzymatic activity can be monitored by spectroscopy at the absorption maximum at 405 nm derived from pNA (Figure 7D). Suspension of the transfected HeLa cells was incubated with *Z*-ALL-pNA, and the endopeptidase activity was then measured (Figure 7E). We found that the enzymatic activity of C143Y mutant-expressing cells was decreased to approximately 20% of that of WT (Figure 7E). Though KIF treatment recovered protein expression of CD10 C143Y and boosted cell surface transport, endopeptidase activity of CD10 was not recuperated (Figure 7E). Of note, the cell permeability of *Z*-ALL-pNA is unclear, and the exact cellular protein expression level of CD10 WT and C143Y is not comparable. Collectively, results from our cellular assay system revealed that KIF retrieves the intracellular transport of CD10 C143Y from the ER to the cell surface but the endopeptidase activity is still inert probably due to conformational imperfection-inappropriate disulfide bonds.

## 3. Discussion

The mechanisms of protein misfolding and its degradation pathway are one of the actual therapeutic targets for conformational diseases. However, understanding the detailed mechanism of this process is complicated because protein misfolding and its clearance are often cell type-dependent machinery [4]. Accumulation of unstable proteins due to inevitable protein misfolding is often found in the group of disorders called conformational diseases. When proteins are misfolded, cellular degradation machineries recognize such toxic precursors and dispose them off rapidly. However, such degradation systems are frequently downregulated by environmental stress, genetic mutation, or protein overproduction. For example, genetic mutation of the alpha1-antitrypsin variant is associated with serious liver sclerosis [4,30]. Cysteine mutation in tyrosinase, which is required for the production of melanin from tyrosine, is linked with albinism and melanoma [31,32].

SCA-related gene mutation produces aberrant products, including polyglutamine-extended proteins, which disturb various cellular functions, and exert cytotoxicity, particularly in the motor neurons. With the recent advances in neurological therapeutics, molecular mechanisms revealed by basic research contribute to the treatment of SCA. In a *Drosophila* model presenting SCA type 6, DNAJ-1, a heat shock protein cofactor, reduces polyglutamine aggregation and neurodegeneration [33]. A guanosine triphosphate derivative facilitates the degradation of polyglutamine aggregates through the ubiquitin–proteasome pathway, and improves the rotor rod performance test by an SCA model mouse [34]. In the present study, we elucidated that the CD10 C143Y mutation found in SCA patients and categorized as SCA43 causes aberrant disulfide bonds formation and downregulation of CD10 functions, such as endopeptidase activity and raft incorporation at the plasma membrane. Furthermore, these malfunctions are strongly associated with the protein turnover pathway in which ERAD machinery targets CD10 C143Y for the proteasomal degradation. We revealed that CD10 C143Y is constitutively degraded through the proteasome and is accumulated to the cytoplasmic aggresome when the proteasome activity is downregulated (Figure 6). The decline in proteasome activity with aging is a well-known aspect [28,35] and strongly links neurodegeneration due to the accumulation of protein aggregates. Therefore, continuous degradation of CD10 C143Y via ERAD might exceed the capacity of the proteasome and could be linked to the deterioration of various cellular functions. We suggest that these phenotypes could be the therapeutic targets for SCA type43 with CD10 cysteine mutation.

For secretory or membrane proteins, disulfide bond formation is one of the important post-translational modifications to maintain protein conformation. Therefore, gene mutation in cysteine residue often causes severe conformational aberrancy and accumulation of misfolded proteins in the ER such as tyrosinase [31,32]. Early research has revealed that disulfide bonds at the C-terminus are important for CD10 protein maturation [29]. CD10 purified from porcine liver under mild conditions exhibited a homodimer through non-covalent bonds [36]. Additionally, according to the crystal structural analysis of human and rabbit CD10 (PDB: 1R1H and 6GID, respectively), a disulfide bond between Cys143 and Cys411 is located near the CD10 dimer interface (Appendix A). This may be the reason for C143Y mutation to induce oligomerization by forming multiple disulfide bonds (Figure 1D). As this mutation also causes failure of endopeptidase activity (Figure 7E), the disulfide bond of C143–C411 may contribute to the enzymatic function and protein conformation of CD10. Maintenance of CD10 activity in vivo is an important factor in Alzheimer’s disease [37,38,39] and cancers [9,40,41]. It has been revealed that, in SH-SY5Y cells, exogenously expressed CD10 downregulates Aβ, which is considered the main cause of Alzheimer’s disease [42]. Despite its importance in clinical treatment, cellular biogenesis of CD10 is not fully understood. Details of the protein maturation and degradation mechanism provide useful insights in understanding the nature of CD10 biogenesis under normal, as well as pathological, conditions, such as tumor development and Alzheimer’s disease.

The ER quality control system recognizes misfolded secretory and membrane proteins to prevent their subsequent transport to the secretory pathway. During the sorting of misfolded proteins for refolding or degradation, BiP, an ER-resident Hsp70 orthologue, retains its client proteins based on the hydrophobicity of the molecular surface. Moreover, BiP determines the fate of its substrates, whether to be refolded or degraded [43,44]. We found that CD10 C143Y is retained in the ER, whereas WT CD10 is transported to the cell surface, and this assortment is probably mediated through BiP association, as revealed by pulldown and subsequent immunoblotting analysis (Figure 3). Reports have revealed that many disease-associated misfolded secretory and membrane proteins, such as low-density lipoprotein receptor [45], transthyretin [46], and thyroglobulin [47], are retained in the ER by BiP association.

In the present study, we identified CD10 C143Y as a novel ERAD substrate that is degraded by representative ERAD components EDEM1, SEL1L, and Hrd1 (Figure 5). EDEM1 possesses mannosidase I-like domain (MLD), which is capable of binding to N-glycan of SEL1L [22,48,49]. EDEM1 accelerates ERAD by handoff of misfolded proteins from EDEM1 to SEL1L [22]. As EDEM1 MLD is required for the association with N-glycan of SEL1L, a mannose analog-type inhibitor KIF reduces the interaction of SEL1L and EDEM1 [22,48]. A recent finding revealed that the free thiol of misfolded ERAD substrates is involved in the recognition process by EDEM1 [50], and the degradation of CD10 C143Y possessing incomplete disulfide bonds may be facilitated by EDEM1. EDEM1 MLD exhibits a substrate interaction property and mannose trimming activity against misfolded ERAD substrates in cultured cells [50]. In our study, as the level of co-expression of EDEM1-HA increased, the remaining CD10 C143Y decreased with faster migration in the immunoblot assay (Figure 5A,B). This band shift accounted for the de-mannosylation activity of EDEM1 MLD toward ERAD substrates [51,52]. Our results support that EDEM1 possesses mannosidase-like activity to facilitate the transport of misfolded protein to the retro-translocation machinery, and this process enhances ERAD of CD10 C143Y.

ERAD machinery contains several E3 ligases destined for poly-ubiquitination and subsequent dislocation to the cytoplasm. Substrate specificity or redundancy is observed during sorting of ERAD substrates. In some cases, one misfolded protein is targeted by several E3 ligases. Thus, selection mechanisms during processing of misfolded proteins by ERAD machinery are not fully provided. For example, degradation of cystic fibrosis transmembrane conductance regulator (CFTR), a multimembrane-spanning chloride channel, is executed by at least seven E3 ligases [53]. While exploring the factors in CD10 C143Y degradation, we found that cytoplasmic E3 ubiquitin ligase C-terminus of Hsp70-interacting protein (CHIP) could enhance the clearance of CD10 C143Y (Appendix A). While this manuscript was in preparation, it was reported that EDEM1 is integrated into the functional ERAD complex, including SEL1L and Hrd1, to facilitate disposal of ER-misfolded proteins through the ERAD pathway or even autophagy [54]. Detailed characterization of the CD10 C143Y quality control and degradation pathway, and the biosynthetic pathway of WT CD10, would offer vital insights for application in SCA treatment, as well as for diseases associated with immature secretory and membrane proteins involved in ER quality control systems. Although our approach using the HeLa cell line is limited to the biochemical molecular aspect, we believe that our results will provide useful information for future translational research and overcoming conformational diseases including SCA.

## 4. Materials and Methods

### 4.1. Antibodies and Chemicals

The antibodies used in the Western blotting experiment were mouse anti-Flag (1:4000; Sigma-Aldrich, St. Louis, MO, USA; F1804), mouse anti-α-tubulin (1:50,000; Sigma-Aldrich, St. Louis, MO, USA; T6074), mouse anti-KDEL (1:4000; Medical and Biological Laboratory, Nagoya, Japan; M181-3), mouse anti-myc (1:1000; Cell Signaling Technology Japan, Tokyo, Japan; #2276), rabbit anti-SEL1L (1:1000; Sigma-Aldrich, St. Louis, MO, USA; S3699), and rat anti-HA (1:1000; Roche Diagnostics K.K., Tokyo, Japan; 11867423001). The antibodies used in the indirect immunostaining experiment were mouse anti-red fluorescent protein (RFP) (1:100; ; Medical and Biological Laboratory, Nagoya, Japan; M155-3) and mouse anti-vimentin (1:200; Santa Cruz biotechnology, Dallas, TX, USA; sc6260). Kifunensine (KIF) was obtained from Enzo Life science (Farmingdale, NY, USA) and MG132 was obtained from Peptide Institute (Osaka, Japan). Chloroquine (Clq) and cycloheximide (CHX) were purchased from Wako chemicals, Osaka, Japan. Anti-Flag M2 agarose beads, E-64, leupeptin, pepstatin A, and aprotinin were purchased from Sigma-Aldrich, St. Louis, MO, USA.

### 4.2. Plasmids

DNA primers used in this study for subcloning are shown in Appendix A. Mouse CD10 expression vector (pSPORT6-CD10) was purchased from DNAform LTD, Yokohama, Japan. Subcloning of CD10 cDNA into pCMV-Flag5a (Invitrogen, Carlsbad, CA, USA) and then into pCX4-bsr vector [55] was carried out by PCR and ligation using In-fusion (Takara Bio, Otsu, Japan). Replacement of Cys143 to Tyr in CD10 was introduced by standard inverse PCR. To construct C-terminal mCherry-tagged CD10, cDNA of wild-type (WT) CD10 and C143Y were subcloned into pmCherry-N1 (Clontech, Mountain View, CA, USA). The C-terminus HA-tagged EDEM1 (EDEM1-HA) expression vector was constructed as follows: cDNA of human EDEM1 from EDEM1-Flag vector [56] was subcloned into pCMV-HA-C vector (Clontech, Mountain View, CA, USA). Mouse myc-tagged WT Hrd1 and E3-dead mutant (c/s) expression vectors [57] were kindly gifted by Dr. Nobuko Hosokawa (Kyoto University, Japan). Plasmid-expressing Sec61βΔTM-SGFP2 by the Tet-on system, as a live cell ER fluorescent marker, was generated and kindly gifted by Dr. Ikuo Wada (Fukushima Medical University, Japan).

### 4.3. Cell Culture and Transfection

HeLa cells, used in several previous studies [58,59] and authenticated using a mycoplasma detection kit (e-Myco^TM^, iNtRON, South Korea), were maintained in DMEM containing 10% fetal bovine serum and 1% penicillin/streptomycin at 37 °C with 5% CO_2_ and humidity. Plasmid transfection was performed using Fugene HD (Roche), according to the manufacturer’s protocol.

### 4.4. Cell Lysis, Protein Turnover Assay, and Western Blotting

After 48 h of transfection, cells were washed once with ice-cold PBS and lysed in a lysis buffer (20 mM HEPES (pH 7.4) containing 100 mM NaCl and 1% Triton X-100) and protease inhibitor cocktail (1 µg/mL of E-64, leupeptin, pepstatin A, and aprotinin) on ice. Detergent-soluble and -insoluble fractions were separated by centrifugation at 18,000× *g* at 4 °C for 10 min. Protein concentration of the soluble fraction was determined by a standard Bradford assay system. Protein expression was analyzed by Western blotting. Denatured proteins in the cell lysate were separated by SDS-PAGE and then transferred to a PVDF membrane (immobilon; Millipore, Burlington, MA, USA). After blocking with 5% skimmed milk in 50 mM Tris-HCl (pH 7.5) containing 100 mM NaCl and 0.05% Tween-20, the membrane was incubated with primary antibodies at room temperature for 1 h. After washing with the above-mentioned Tris buffer without skimmed milk, the membranes were treated with HRP-conjugated secondary antibodies. Protein bands were detected using ECL reagents and the Chemidoc system (Bio-Rad, Hercules, CA, USA). Densitometric analysis was performed using the ImageJ software version 1.50 (National Institute of Health, USA). For protein turnover assay, transfected cells were treated with 10 µg/mL cycloheximide (CHX) to suppress translation for indicated times. To examine the degradation process, CHX-chase was performed in the presence or absence of protein degradation inhibitors.

### 4.5. Glycosidase Digestion

For N-glycan processing analysis, cell lysate prepared as described above was mock-treated or incubated with endoglycosidase H (Endo H) or PNGaseF, purchased from New England BioLabs (Ipswich, MA, USA) according to the manufacturer’s protocol.

### 4.6. RNA Interference (RNAi) Experiments

The targeted sequences were as follows: 5′-CGU ACG CGG AAU ACU UCG AdTdT-3′, (purchased from Japan Bio Services Co LTD, Asaka, Japan) or human SEL1L: 5′-UUA ACU UGA ACU CCU CUC CCA UAG A-3′ (kindly gifted by Dr. Nobuko Hosokawa, Kyoto Univ. [60]). RNAi oligos were introduced into HeLa cells using RNAiMAX (Invitrogen, Carlsbad, CA, USA) at 10 nM and incubated at 37 ℃ for 72 h.

### 4.7. CD10 Endopeptidase Assay

HeLa cells transfected with empty or CD10-expressing plasmids for 48 h were collected using a cell scraper and suspended in PBS containing 10 µM ZnCl_2_ and the protease inhibitor cocktail described above. *Z*-Ala-Leu-Leu-*p*-nitroanilide (ZALL-*p*NA; Peptide institute, Osaka, Japan) was added to the cell suspension at the final concentration of 250 µM and incubated at 37 °C for 1 h. After the reaction, the relative enzymatic activity was measured by determining the absorption maximum of isolated *p-*nitroanilide (Figure 7D) at 405 nm using a spectrometer. The number of HeLa cells transfected with the empty vector was used as the cellular background and was subtracted from each CD10-expressing cellular sample.

### 4.8. Immunoisolation of Flag-Tagged Proteins

Transfected cells were washed once with PBS and lysed in PBS containing 1% NP-40 and the protease inhibitor cocktail, on ice. The detergent-soluble fraction was incubated with anti-Flag M2 agarose beads at 4 °C for 1 h with gentle rotation, followed by centrifugation at 3000 rpm for 10 min at 4 °C. The precipitated beads were washed four times with 50 mM Tris-HCl (pH 7.5) containing 150 mM NaCl and 0.1% 3-[(3-cholamidopropyl) dimethylammonio] propanesulfonate (CHAPS). Flag-tagged proteins were eluted by incubation with 500 µM Flag peptide (Sigma-Aldrich, St. Louis, MO, USA) in 50 mM Tris-HCl (pH 7.5) containing 150 mM NaCl at room temperature for 15 min with gentle rotation. Extracted proteins were analyzed by Western blotting as described above.

### 4.9. Indirect Immunostaining

The indirect immunostaining method was based on our previous research [61]. After transfection, cells plated onto the coverslip were fixed with 4% paraformaldehyde in PBS (Wako chemicals, Osaka, Japan) at room temperature for 10 min. Membrane permeabilization was conducted using 0.1% Triton X-100 in immunostaining buffer (PBS containing 5% glycerol and 1% goat serum) at 2 °C for 1 min and then blocked with the above-mentioned buffer at room temperature for 5 min. This permeabilization process was omitted in the detection of cell surface proteins. Cells were incubated with primary antibody (mouse anti-RFP or anti-vimentin) and then stained with goat secondary antibody conjugated with Alexa488 or 594 (Molecular Probes, Eugene, OR, USA). After rinsing with distilled water, coverslips were mounted face-down on slide glasses with Mowiol (Sigma-Aldrich, St. Louis, MO, USA). Images were captured by a confocal laser microscope with an oil immersion objective (LSM 780, Carl Zeiss, Inc., Oberkochen, Germany) under a magnification of 63×. For live cell analysis, HeLa cells were cultured in a glass-bottom dish (Wako chemicals, Osaka, Japan). After transfection and drug treatment, cell culture media were replaced with a live imaging solution (Thermo Fisher Scientific, Waltham, MA, USA) and confocal microscopic analysis was performed on a heated stage (Tokai Hit, Fujinomiya, Japan) at 37 °C.

## Figures and Tables

**Figure 1 ijms-21-04237-f001:**
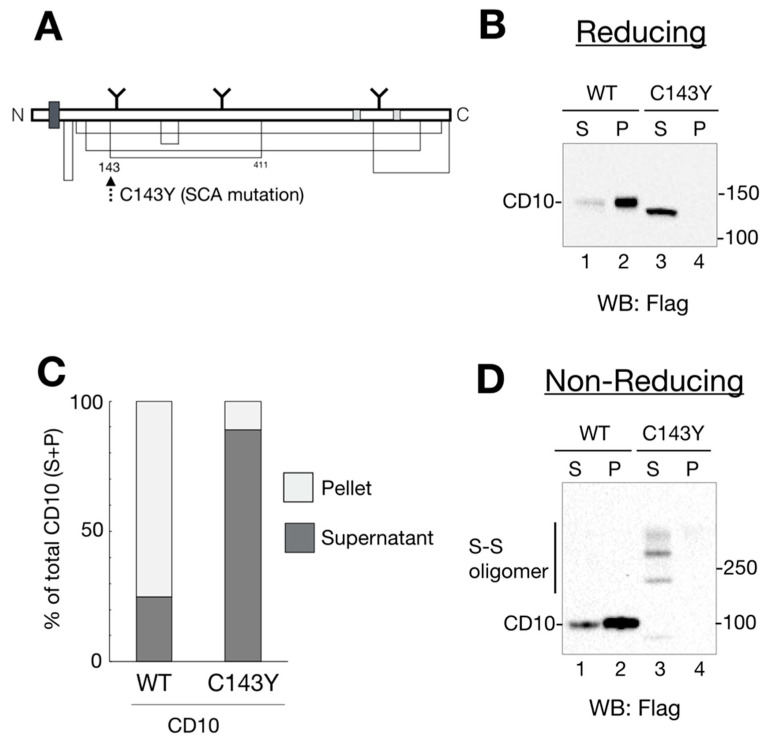
Spinocerebellar ataxia (SCA)-derived CD10 cysteine mutation causes aberrant disulfide bond formation and insufficient raft localization. (**A**) schematic representation of human CD10 structure. The thick dark gray square shows the transmembrane domain. N-linked glycans are noted as Y motifs with the glycosylation position. Enzyme catalytic sites are depicted in light gray. Disulfide bonds are shown as a solid line. Location of a SCA-related C143Y mutation is denoted by a dotted arrow. (**B**) wildtype (WT) CD10 is detected in a Triton X-100-insoluble fraction, whereas C143Y is solubilized. HeLa cells transfected with WT CD10 or C143Y-Flag vector were lysed and fractionated into detergent-soluble (S; soluble) and -insoluble (P; pellet) fractions. Proteins were resolved in reducing SDS-PAGE, followed by immunoblotting using anti-Flag antibody. Molecular weight of the marker proteins is shown on the right side of the gel. Lane numbers are shown on the lower side of the gel. (**C**) Quantification of B (ratio of S and P) is shown. The percentage of each band (WT, lanes 1 and 2; C143Y, lanes 3 and 4) is relative to the sum of all bands (100%). S and P are represented as dark gray and light gray, respectively. Results are shown as the ratio of proteins to the total protein (100%). Values are means of three independent experiments. (**D**) C143Y mutation induces disulfide bond-mediated oligomerization. Samples were prepared as mentioned in B, except for a denaturing process under the non-reducing condition. CD10 oligomers with disulfide bonds are denoted as “S–S oligomer” on the left side of the gel. Molecular weight of the marker proteins is shown on the right side of the gel.

**Figure 2 ijms-21-04237-f002:**
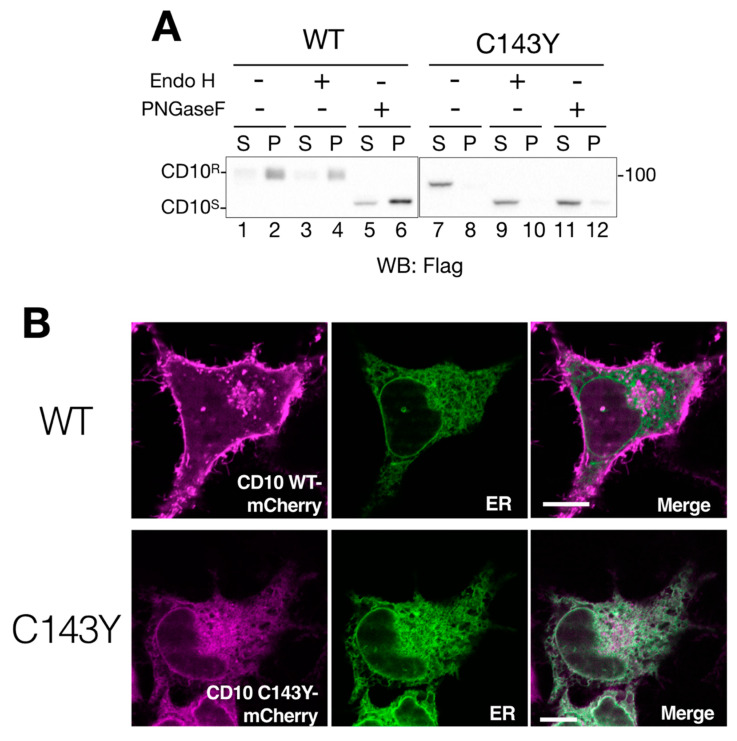
Cell surface expression of CD10 is impeded by C143Y mutation. (**A**) imperfect cellular transport of CD10 C143Y. HeLa cells were transfected with C-terminus Flag-tagged WT CD10 or C143Y for 48 h. After cell lysis and reducing SDS-PAGE, WT CD10 (lanes 1–6) and mutant C143Y (lane 7–12) were visualized by immunoblotting using anti-Flag antibody. Denatured cell lysates were mock-treated (lanes 1–2 and 7–8, respectively) and digested with endo H (lanes 3–4 and 9–10) or PNGase F (5–6 and 11–12, respectively). The endoglycosidase H (Endo H)-resistant form (CD10^R^) and glycosidase-sensitive form (CD10^S^) are denoted on the left side of the gel. Molecular weight markers are shown on the right side of the gel. (**B**) CD10 C143Y is located at the endoplasmic reticulum (ER). HeLa cells cultured on a glass bottom dish were transfected with C-terminus mCherry-tagged WT CD10 or C143Y and Sec61βΔTM-SGFP2 (as an ER marker). Fluorescent images were observed using confocal microscopy in the live cell condition. Scale bars represent 10 μm.

**Figure 3 ijms-21-04237-f003:**
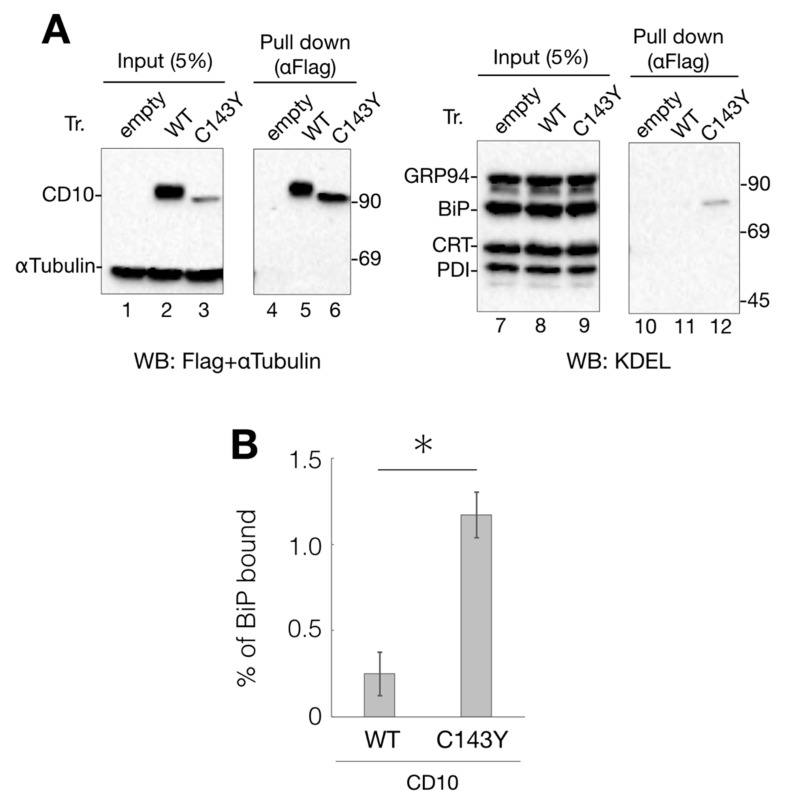
Interaction of CD10 C143Y with ER chaperone BiP. (**A**) association of BiP with CD10 C143Y. Flag-tagged WT CD10 and C143Y expressed in HeLa cells were extracted with 1% NP-40-containing buffer. Lysate (5%) was loaded as the input (lanes 1–3 and 7–9, respectively). WT CD10 and C143Y-Flag were immunoisolated using anti-Flag agarose beads and eluted with Flag peptide (lane 4–6 and 10–12, respectively). Extracted proteins were resolved by reducing SDS-PAGE and followed by immunoblotting with indicated antibodies. Molecular weight markers are shown on the right side of the gel. (**B**) percentage of WT CD10 or C143Y-Flag bound to BiP was calculated as follows: ((BiP proteins after anti-Flag immunoprecipitation)/(BiP of cell lysates) × 20) × 100. Error bars represent standard deviations for three independent experiments. Asterisk indicates statistical significance (*p* < 0.05) by Student’s *t* test.

**Figure 4 ijms-21-04237-f004:**
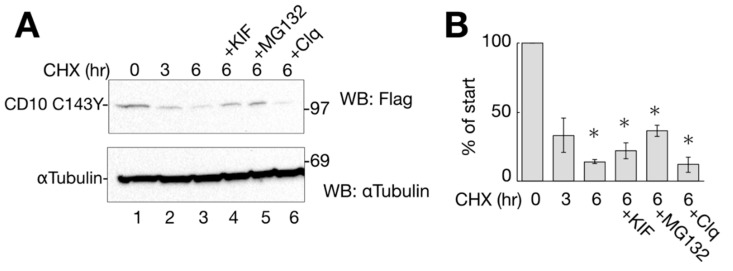
CD10 C143Y turned over shortly. (**A**) degradation of CD10 C143Y. HeLa cells transfected with CD10 C143Y-Flag were treated with cycloheximide (CHX) for indicated times with or without degradation inhibitors (kifunensine (KIF), MG132, or Clq; lanes 4–6). Upon treatment, cells were harvested, and immunoblotting with the indicated antibodies was performed. α-tubulin was used as the cellular loading control. (**B**) half-life of CD10 C143Y-Flag was measured for densitometric analysis of A. CD10 C143Y-Flag bands were normalized to that of α-tubulin as the total cellular protein. Asterisk refers to time 0 h (*p* < 0.01) by Student’s *t* test. Error bars represent standard deviations for at least three independent experiments.

**Figure 5 ijms-21-04237-f005:**
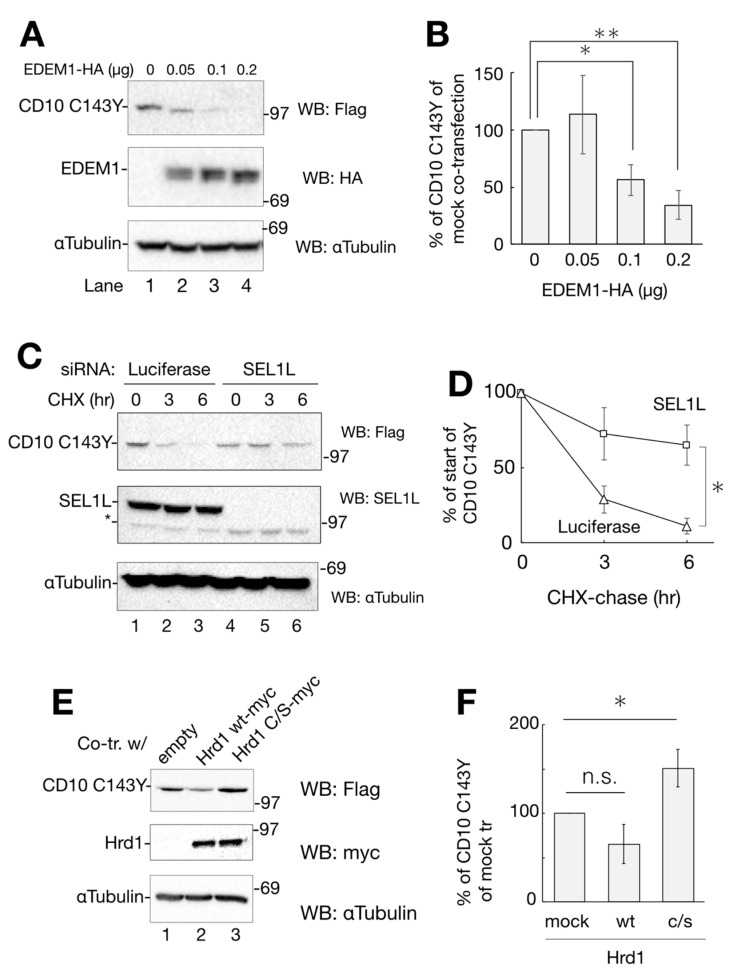
Degradation of CD10 C143Y through ER-associated degradation (ERAD). (**A**) overexpression of EDEM1-HA enhances CD10 C143Y clearance. Co-transfected HeLa cells with CD10 C143Y-Flag and EDEM1-HA (lanes 1–4; 0, 0.1, 0.15, 0.2 μg, respectively) were subjected to Western blotting with the indicated antibodies. (**B**) data in A were quantified. Relative CD10 C143Y-Flag signals to mock transfection are plotted. Asterisk refers to 0 μg of EDEM1-HA plasmid, indicating statistical significance (*, *p* < 0.05: **, *p* < 0.01) by Student’s *t* test. (**C**) SEL1L is required for CD10 C143Y degradation. HeLa cells were transfected with siRNA to luciferase (non-target control) or SEL1L, followed by transfection with CD10 C143Y-Flag. After CHX-chase, cellular proteins were resolved and detected by immunoblotting with the indicated antibodies. Asterisk indicates the non-specific band derived from the SEL1L antibody. (**D**) quantification of C. Values of CD10 C143Y-Flag under siRNA to SEL1L (squares) or luciferase (triangles) are shown. Error bars represents standard deviation for at least three independent experiments. Asterisk at the time point of 6 h indicates statistical significance (*p* < 0.05) by Student’s *t* test. (**E**) enzymatic activity of Hrd1 is important for CD10 C143Y clearance. CD10 C143Y-Flag was introduced into HeLa cells with an empty vector, Hrd1 WT-myc or Hrd1 c/s-myc. Protein expression level was analyzed by Western blotting with the indicated antibodies. (**F**) quantification of E. Asterisk indicates statistical significance (*p* < 0.05) by Student’s *t* test, and n.s. indicates non-significance (*p* = 0.072). In all densitometric analyses, error bars represent the means ± standard deviation for at least three independent experiments.

**Figure 6 ijms-21-04237-f006:**
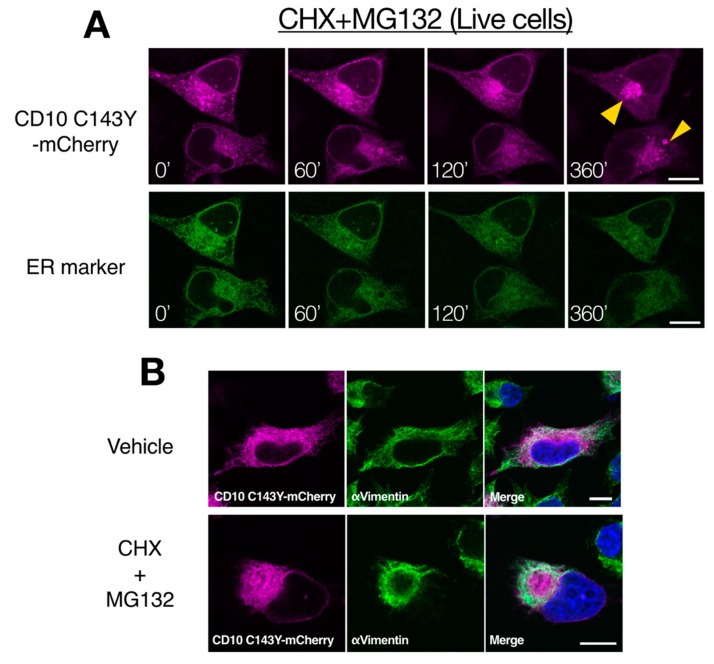
Proteasome inactivation causes accumulation of CD10 C143Y at the cytoplasmic aggresome. (**A**) retro-translocation of CD10 C143Y. CD10 C143Y-mCherry was tracked during inactivation of the proteasome. Live cell imaging was conducted for HeLa cells transfected with CD10 C143Y-mCherry and Sec62ΔTM-SCGFP2 (ER marker) on a glass bottom dish. Cells were treated with CHX and MG132, and time-lapse images (0–360 min) were captured using a confocal microscope. Scale bars represent 10 μm. (**B**) accumulation of CD10 C143Y at the aggresome. HeLa cells were transfected with CD10 C143Y-mCherry and incubated on the coverslip. Before fixation, cells were treated with or without CHX and MG132 for 6 h. Indirect immunostaining with anti-vimentin antibody was performed, and images were captured using a confocal microscopic analysis. Scale bars represent 10 μm.

**Figure 7 ijms-21-04237-f007:**
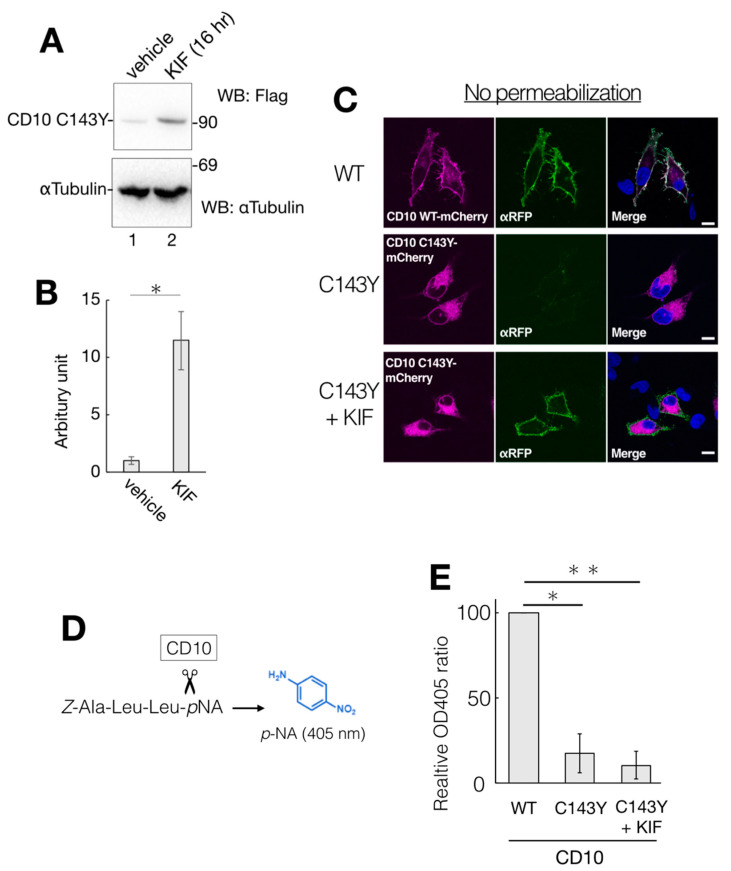
Partial recovery of CD10 C143Y by KIF treatment. (**A**) HeLa cells-expressed CD10 C143Y were treated with vehicle or 100 µM KIF for the last 16 h of total transfection time. After lysis, supernatants of cell lysates were resolved and immunoblotted with anti-Flag or anti-αTubulin. (**B**) quantification of anti-Flag immunoblotting from A. Error bars represent mean ± standard deviation for three independent experiments. Asterisks indicate statistical significance (*; *p* < 0.01) by Student’s *t* test. (**C**) recovery of cell surface expression of CD10 C143Y by KIF treatment. Transfected HeLa cells with mCherry-tagged WT or C143Y CD10 plasmids with or without KIF were fixed with PFA and immunostained with anti-red fluorescent protein (RFP) antibody without plasma membrane permeabilization. Scale bars represent 10 μm. (**D**) overview of endopeptidase assay of CD10. A peptide substrate *Z*-Ala-Leu-Leu-*p*NA was cleaved next to the second Leu by CD10. The enzyme activity was measured by liberated *p*NA, which exhibits an absorption maximum at 405 nm. (**E**) C143Y showed less enzymatic activity than WT even under 16 h KIF treatment. Transfected HeLa cells with indicated vectors were subjected to CD10 enzyme activity assay, as described in Materials and Methods. Error bars represent mean ± standard deviation for three independent experiments. Asterisks indicate statistical significance (*; *p* < 0.01, **; *p* < 0.01) by Student’s *t* test.

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
