# Peer review of "Endoplasmic Reticulum Associated Degradation of Spinocerebellar Ataxia-Related CD10 Cysteine Mutant"

_ijms, 2020, doi:10.3390/ijms21124237_

Round 1
Reviewer 1 Report
The paper by Manuka et al. entitled “ Endoplasmic Reticulum associated degradation of Spinocerebellar ataxia-related CD10 cysteine mutant” presents that CD10 C143Y is a novel ERAD substrate, and provides the molecular basis for understanding SCA due to cysteine mutation in CD10. It is highly interesting study which with potential clinical implications.
Introduction is coherent and informative. Results are clearly described and Figures are of good quality with detailed description. Discussion is well prepared. Methods are sufficiently described. References are novel and in the proper style.
My questions were listed below:
- The authors should explain why they used HeLa cells in their experiments and not e.g. primary brain cells? Could there be a difference in brain cells?
- Limitation to the study should be added at the end of discussion part as well as concluding sentence.
Reviewer 2 Report
In this manuscript Kanula et al. describe in vitro how a Spinocerebellar ataxia –related neprylisin mutant may evade the normal intracellular degradation pathways thus contributing to the development of the disease. The rationale behind this idea is sound and the experimental approach adopted to support the main hypothesis was designed accurately. Next, all experiments are described in detail and consistently supported the main conclusions of the work.
However, there are some issues that should be taken into account before considering this manuscript acceptable for publication.
1) CD10 or nerprylisin (NEP) plays a fundamental role in amyloid beta peptide processing. Have the authors addressed the capacity of the CD mutant to degrade Abeta? This issue could significantly contribute to further shed light in the neurodegeneration mechanisms linked to aberrant CD mutations.
2) The authors propose that mutation in Cys143 residue may cause incorrect disulfide bond formation which is, in turn, sensitive to reducing conditions. However, proteasomes are also highly sensitive to the presence of reducing conditions and this fact may influence the interplay between redox stress, proteasome activity and CD10 trafficking and degradtion (a useful reading is: Coordination Chemistry Reviews 347, 1-22, 2017). I suggest that this important issue should be mentioned in the discussion and related papers accurately referenced.
minor points:
Abstract:
The sentence: …. While CD10 C143Y forms a disulfide bond-mediated oligomer, leading to significantly reduced enzymatic activity… is not fully clear to me. Please, rephrase it.
Introduction: line 4 .. replace “proteinaceous” with “proteolytic” and “human homeostasis” with “cell homeostasis”
